# Isolation and Characterization of Contemporary Bovine Coronavirus Strains

**DOI:** 10.3390/v16060965

**Published:** 2024-06-16

**Authors:** Yu Li, Roberto A. Palomares, Mingde Liu, Jiayu Xu, Chohee Koo, Francesca Granberry, Samantha R. Locke, Greg Habing, Linda J. Saif, Leyi Wang, Qiuhong Wang

**Affiliations:** 1Center for Food Animal Health, Department of Animal Sciences, College of Food, Agricultural and Environmental Sciences, The Ohio State University, Wooster, OH 44691, USA; li.13226@osu.edu (Y.L.); liu.6202@osu.edu (M.L.); xu.4008@osu.edu (J.X.); saif.2@osu.edu (L.J.S.); 2Department of Population Health, College of Veterinary Medicine, University of Georgia, 2200 College Station Rd., Athens, GA 30602, USA; palomnr@uga.edu (R.A.P.); chohee.koo@uga.edu (C.K.); francesca.granberry@uga.edu (F.G.); 3Department of Veterinary Preventive Medicine, College of Veterinary Medicine, The Ohio State University, Columbus, OH 43210, USA; locke.91@osu.edu (S.R.L.); habing.4@osu.edu (G.H.); 4Veterinary Diagnostic Laboratory, Department of Veterinary Clinical Medicine, College of Veterinary Medicine, University of Illinois, Urbana, IL 61802, USA

**Keywords:** bovine coronavirus, virus isolation, genomic sequence

## Abstract

Bovine coronavirus (BCoV) poses a threat to cattle health worldwide, contributing to both respiratory and enteric diseases. However, few contemporary strains have been isolated. In this study, 71 samples (10 nasal and 61 fecal) were collected from one farm in Ohio in 2021 and three farms in Georgia in 2023. They were screened by BCoV-specific real-time reverse transcription-PCR, and 15 BCoV-positive samples were identified. Among them, five BCoV strains from fecal samples were isolated using human rectal tumor-18 (HRT-18) cells. The genomic sequences of five strains were obtained. The phylogenetic analysis illustrated that these new strains clustered with US BCoVs that have been detected since the 1990s. Sequence analyses of the spike proteins of four pairs of BCoVs, with each pair originally collected from the respiratory and enteric sites of one animal, revealed the potential amino acid residue patterns, such as D1180 for all four enteric BCoVs and G1180 for three of four respiratory BCoVs. This project provides new BCoV isolates and sequences and underscores the genetic diversity of BcoVs, the unknown mechanisms of disease types, and the necessity of sustained surveillance and research for BCoVs.

## 1. Introduction

Bovine coronavirus (BCoV) belongs to the *Embecovirus* subgenus of the *Betacoronavirus* genus within the *Orthocoronavirinae* subfamily of the *Coronaviridae* family. Betacoronaviruses, along with alphacoronaviruses, infect exclusively mammals, whereas gammacoronaviruses and deltacoronaviruses primarily infect avian species [1]. BCoV was initially identified by Mebus et al., who discovered coronavirus-like particles in a diarrheic calf’s feces by electron microscopy in Nebraska in 1972 [2,3]. Subsequent experiments confirmed that the BCoV found in the fecal material caused diarrhea in infected calves and was closely related to bovine-like CoVs in other mammals such as dogs, sheep, and deer [3,4,5,6,7]. BCoV viral particles are enveloped and pleomorphic with a diameter of around 65–210 nm and spike (S) and hemagglutinin-esterase (HE) protein projections [4,8,9]. The BCoV genome possesses open reading frame (ORF) 1a, ORF1b, and the ORFs for the structural and accessory proteins, including the 32 kDa, HE, S, 4.9 kDa, 4.8 kDa, envelope (E), membrane (M), and nucleocapsid (N) proteins [10,11,12]. BCoV possesses the unique HE gene, which is absent from other subgenera of betacoronaviruses, alphacoronaviruses, gammacoronaviruses, and deltacoronaviruses.

BCoV infects both the upper and lower respiratory and gastrointestinal tracts of cattle, being shed in feces and nasal secretions [4,8]. Coronaviruses undergo continuous evolution through the accumulation of point mutations, insertions and deletions, and recombination events, resulting in the emergence of new variants capable of evading host immune responses [13]. Despite being half a century since the discovery of the first BCoV, significant knowledge gaps exist regarding BCoV pathogenesis and evolution.

The S protein of coronaviruses serves as the receptor-binding protein and is a major target for inducing protective immunity. It shows significant genetic variations among strains due to immune pressure. Changes in tissue tropism among coronaviruses often correlate with mutations in the S protein. For instance, the deletion of the N-terminal domain of the S protein, and sometimes together with deletions in ORF3a and ORF3b of transmissible gastroenteritis virus (TGEV), an alphacoronavirus, shifted the viral major tropism from enteric to the respiratory tract, resulting in porcine respiratory coronavirus (PRCV) [14,15,16,17]. Therefore, a comprehensive investigation of BCoV S protein sequence variation and domain function is essential for the understanding of the virus adaptation and transmission mechanisms.

Furthermore, BCoV serves as a valuable model for molecular studies of coronavirus. Critical questions, such as the mechanisms underlying the manifestation of enteric or respiratory diseases, and the role of the HE protein, remain unanswered. Thus, gaining insights into the current BCoV genomes and the isolation of contemporary strains are imperative steps toward establishing a reverse genetics platform in our laboratory.

In this study, we detected and isolated contemporary BCoVs from cattle samples. Then, we performed sequence analyses of BCoVs at the genomic, S protein, and HE protein levels. Additionally, by comparing four pairs of fecal and nasal samples, we found potential associations between the virus strain tissue origin and viral genetic markers. These results provide insights into the BCoV strain genetic diversity and molecular characteristics.

## 2. Materials and Methods

### 2.1. Sample Collection, Screening, and Preparation

Sample collection information is listed in Table 1. Ten pairs of fecal and nasal swab samples were collected in March 2021 from 10 veal calves, which were separated from their mother and supplemented with milk and had no apparent clinical signs on a veal farm in Ohio (OH). An additional 51 samples were obtained in April–May 2023 from various age groups of cattle, including pre-weaned calves (n = 27), nursing beef (n = 1), weaned feeder calves (n = 20), and unknown-age cattle (n = 3) of Holstein, Jersey, Nigerian Red, and Angus breeds from three farms in Georgia (GA1, GA2, and GA3). Samples were diluted tenfold with phosphate-buffered saline without Mg2+ and Ca2+ [PBS(−)] (Gibco, Carlsbad, CA, USA), vortexed, and then centrifuged at 2000× *g* at 4 °C for 10 min. The supernatant was subsequently filtered through 0.22 μm pore size filters (Millipore, Chicago, IL, USA) to remove bacteria. The filtered samples were then stored at −80 °C until further analysis. Diluted samples underwent TaqMan real-time reverse transcription-PCR (RT-qPCR) assay targeting the conserved M gene of BCoV for the detection of viral RNA titers [18].

### 2.2. Virus Isolation and Propagation

The cloned human ileocecal colorectal human rectal tumor-18 (HRT-18) cell line (provided by Dr. Linda Saif, The Ohio State University, Wooster, OH, USA) was selected for the isolation of BCoVs from the RT-qPCR positive samples from Ohio and Georgia [19]. The growth medium (GM) for HRT-18 cells consisted of advanced minimum essential medium (AMEM) (Gibco, Carlsbad, CA, USA), 1% antibiotic–antimycotic (Gibco, Carlsbad, CA, USA), 1% l-glutamine (Gibco, Carlsbad, CA, USA), and 10% heat-inactivated fetal bovine serum (FBS, Hyclone, Logan, UT, USA).

Two-day-old HRT-18 cells reaching two-thirds confluency in 24-well plates were used for virus inoculation. First, the GM was replaced with AMEM and incubated for 1 h at 37 °C. Then, the cell monolayers were inoculated with 200 μL per well of diluted sample, with one sample being added to four wells. After incubating the plate for 60 min at 37 °C with 5% CO_2_, the supernatant was removed from two wells followed by gentle washing using PBS(−) twice [Inoc (−) condition], while the supernatant was retained in the other two wells [Inoc (+) condition]. AMEM supplemented with 5 μg/mL of trypsin (Gibco, Carlsbad, CA, USA) was added to all wells. At this stage, 50 μL of supernatant was collected from each well for RNA extraction to establish a baseline at 1 h post-inoculation (hpi) for RNA measurement.

Cytopathogenic effects (CPEs) were monitored in the following days. When either 80% CPE was observed, or at four days post-inoculation (dpi) without CPE, plates were frozen at −80 °C and thawed once. All harvested supernatants were subjected to RNA extraction and tested for BCoV using RT-qPCR to calculate the difference in cycle threshold (ΔCt) values between the samples at 1 hpi and harvested time.

Samples with ΔCt > 1 (n = 15) were subjected to serial passages, with the first three passages conducted in the same manner to confirm the successful isolation of the virus in HRT-18 cells. In serial passaging, the procedure was modified to utilize two wells per sample, with 200 μL of harvested supernatant inoculated per well in 12-well plates. After a 60-minute incubation in a 5% CO_2_ incubator at 37 °C, the inoculum was removed, and AMEM with trypsin was added directly.

### 2.3. Viral RNA Extraction

Viral RNA was extracted from 50 μL of the field sample suspension and the culture supernatants using the 5× MagMAX-96 Viral Kit (Invitrogen, Carlsbad, CA, USA) and the MagMax™ Express machine (Applied Biosystems, Bedford, MA, USA), following the manufacturer’s instructions. Finally, 50 μL of RNA was obtained in an elution buffer.

### 2.4. TaqMan RT-qPCR and Conventional PCR

For the RT-qPCR targeting the conserved M gene, the previously reported forward primer (5′-CTGGAAGTTGGTGGAGTT-3′), reverse primer (5′-ATTATCGGCCTAACATACATC-3′), and probe (6-carboxyfluorescein-CCTTCATATCTATACACATCAAGTTGTT-black hole quencher 1) [18] were synthesized by Integrated DNA Technologies (https://www.idtdna.com/pages, accessed on 25 April 2024), and the reaction system was implemented using 2 μL of RNA and the Qiagen OneStep RT-PCR kit (Qiagen, Valencia, CA, USA) on RealPlex real-time thermocyclers (Eppendorf, Barkhausenweg, Hamburg, Germany). Both viral genomic and several subgenomic RNAs that contain the M gene are detected by this assay. The whole genome sequence (WGS) of the BC18 strain was obtained using conventional PCR with multiple primers (shown in Appendix A) followed by Sanger sequencing at the Comprehensive Cancer Center of The Ohio State University or Oxford nanopore sequencing at Plasmidsaurus (https://www.plasmidsaurus.com/, accessed on 25 April 2024). Initially, viral RNA was reverse transcribed to cDNA using the SuperScript™ IV cDNA synthesis kit (Invitrogen, Carlsbad, CA, USA) employing a strategy combining oligo(dT) and random hexamers’ priming. Subsequently, the cDNA was amplified with the BCoV-specific primers using the high-fidelity PrimeSTAR GXL DNA polymerase (TaKaRa, Toshima-ku, Tokyo, Japan). The PCR reaction mixture (50 μL) consisted of 2 μL of cDNA, 10 μL of 5× PCR buffer, 4 μL of deoxynucleotide triphosphates (dNTPs) (2.5 mM each), 2 μL of GXL PCR enzyme, and 1 μL each of forward and reverse primers (10 µM each). The PCR reaction followed thermal cycling conditions: 98 °C for 30 s, 30 cycles of 98 °C for 10 s, 55 °C/60 °C for 15 s, and 68 °C for 1–3 min, with a final extension step at 68 °C for 5 min. PCR products were analyzed using 1% agarose gel electrophoresis to confirm the proper sizes of amplicons. Subsequently, PCR products of the correct size were purified using the QIAquick Gel extraction kit (Qiagen, Valencia, CA, USA) for subsequent sequencing.

### 2.5. Immunofluorescence Assay (IFA)

BCoV-infected HRT-18 cells in 6-well plates were fixed and permeabilized with 100% methanol at −20 °C for 15 min. Subsequently, the cells were washed five times with PBS and then incubated with 5% bovine serum albumin (BSA) at room temperature for one hour to block nonspecific binding sites. The primary antibody, guinea pig hyperimmune antiserum to the Mebus strain of BCoV, NR455 (diluted 1:2000) (provided by Dr. Linda Saif, The Ohio State University), was added to the fixed cells and incubated at room temperature for one hour. After washing, the secondary antibody, fluorescein AF488-conjugated goat anti-guinea pig IgG (diluted 1:1000) (Invitrogen, Carlsbad, CA, USA), was applied and incubated at room temperature for one hour in the dark. After incubation, the cells were washed five times with PBS. Subsequently, staining of the nuclei was performed using 4′,6-diamidino-2-phenylindole (DAPI). The plates were then observed using an IX-70 fluorescence microscope (Olympus, Center Valley, PA, USA).

### 2.6. Immunoelectron Microscopy (IEM)

For sample preparation, BCoV BC8-infected HRT-18 cell cultures were collected at 48 hpi and centrifuged at 2095× *g* for 30 min at 4 °C, and supernatants were collected. The guinea pig hyperimmune antiserum NR455 was diluted 1:1000 in PBS, ultracentrifuged at 17,000× *g* for 30 min at 4 °C using SW41 Ti rotor (Beckman Coulter, Brea, CA, USA), and filtered through a 0.22 µm filter before incubation with the samples. Cell culture supernatants were filtered through a 0.45 µm filter and then mixed with the diluted antiserum with gentle shaking at 4 °C for 12–18 h. The samples were subsequently ultracentrifuged at 14,000× *g* at 4 °C for 1.5 h using the SW32 Ti rotor, washed once with AMEM, and then ultracentrifuged again. To detect virion particles, samples were stained with an equal volume of uranyl acetate (2% in ddH_2_O), incubated for 1 min, and applied to a 300 mesh formvar-coated copper grid. After incubating on the grid for 3–5 min, the excess sample was absorbed using filter paper, and the viral particles were visualized using an H7500 electron microscope (Hitachi High Technologies, Minato-ku, Tokyo, Japan).

### 2.7. Plaque Assay for BCoV Titration and Purification of the BC8 Strain

HRT-18 cells were seeded in 6-well plates, reaching semi-confluency on the first day. Once the cells reached 100% confluency, the growth medium (GM) was replaced with AMEM and incubated for one hour at 37 °C. Subsequently, the AMEM was removed, and the cell monolayers were inoculated with a 10-fold serially diluted and well-vortexed sample (500 μL/well), with duplicates per dilution. Plates were gently shaken every 15 min. After one hour of incubation, the inoculum was removed, and the cell monolayer was washed twice with PBS(−). A mixture of equal volumes of 1.5% SeaPlaque agarose (Lonza, Walkersville, MD, USA) and 2× MEM containing 2% antibiotic–antimycotic, 2% glutamine, and 10 µg/mL trypsin was prepared. The 0.75% agarose mixture was poured onto the cell monolayers, and the plates were kept in a hood for approximately 20 min until the agarose solidified. Subsequently, the plates were inverted and incubated at 37 °C for a maximum of five days, with staining occurring on the last day.

For the titration of a sample, on the fourth day, the cells were fixed using 10% neutral buffered formalin for 15 min, followed by staining with 1% crystal violet (Invitrogen, Carlsbad, CA, USA) for one minute to visualize obvious plaques. Virus titers in plaque forming units (PFUs)/mL were calculated based on the last two dilutions that showed plaques. For plaque purification, the culture was directly stained with 0.33% neutral red (Invitrogen, Carlsbad, CA, USA), with incubation for a maximum of three hours. After incubation, the dye was removed, and the plaques were visualized. Individual plaques were confirmed using an LED light box and an optical microscope (Olympus, Hachioji-shi, Tokyo, Japan). Individual plaques were picked using sterile pipette tips and transferred to 500 μL of AMEM. The plaque purification process was repeated at least twice for each isolated virus strain to ensure virus purity.

### 2.8. Growth Kinetics and Plaque Size Calculation

HRT-18 cells were inoculated with BCoV at a multiplicity of infection (MOI) of 0.01 and incubated for one hour at 37 °C. The culture supernatants were then collected at various time points post-infection: 1 hpi, 12 hpi, 24 hpi, 36 hpi, 48 hpi, 60 hpi, 72 hpi, 84 hpi, and 96 hpi, respectively. Infectious viral titers were determined by plaque assay. The growth kinetics of one of the newly isolated BCoV strains, BC8, were compared with those of the historical DBA BCoV strain [20]. The diameter of 30 plaques was measured for each strain to calculate plaque size.

### 2.9. Sequencing, Protein Structure Prediction, and Phylogenetic Analysis

We isolated five strains: BC7, BC8, BC9, BC39, and BC47. Because BC7 was from the same farm (Ohio) as BC8 and BC9, we just sent BC8 and BC9 for whole genome sequencing using next-generation sequencing (NGS) at the University of Illinois at Urbana-Champaign. We chose the strain BC8 at passage 2 (BC8-P2), BC9-P3, BC39-P5, and BC47-P3 because the original fecal samples had low viral RNA titers (with a Ct > 26) and reached high titers after passaging in HRT-18 cells (with a Ct < 25). Nucleic acids of these isolates were extracted on KingFisher Flex using the MagMAX™ Pathogen RNA/DNA Kit (ThermoFisher, Waltham, MA, USA), following the kit manual. Nucleic acid samples were subject to sequence-independent, single-primer amplification (SISPA). In detail, the nucleic acids were reverse transcribed into cDNA using Superscript III (ThermoFisher, Waltham, MA, USA) and a random octamer primer (GACCATCTAGCGACCTCCACNNNNNNNN), converted into dsDNA by Klenow polymerase (NEB, Ipswich, MA, USA), and further amplified using a single primer (GACCATCTAGCGACCTCCAC) and the Advantage 2 PCR kit (Takara Bio, Ann Arbor, MI, USA). The PCR products were purified using the QIAquick PCR Purification Kit (QIAGEN, Germantown, MD, USA) and quantified using Qubit broad-range and high-sensitivity kits (ThermoFisher, Waltham, MA, USA). The NGS library of each sample was prepared using the Nextera XT kit (Illumina, San Diego, CA, USA) and sequenced on Illumina MiSeq. FASTQ files of each isolate were assembled using SPAdes version v3.14.0 and the assembled sequences were blasted against the NCBI NT local blast database. For strain BC18, WGS was acquired using conventional PCR with the primers shown in Appendix A. Purified PCR products were sent to PlasmidSaurus (https://www.plasmidsaurus.com/, accessed on 25 April 2024) to obtain lineage DNA sequences, and overlapping DNA sequences were then analyzed to assemble the complete WGS.

Sequence alignments of the S proteins and HE proteins of three historical pairs (EF424615 and EF424617, AF391541 and AF391542, and EF424619, and EF424620) and one pair (OR502440 and OR502442) from this study of respiratory and enteric strains were performed using Clustal Omega (https://www.ebi.ac.uk/Tools/msa/clustalo/, accessed on 25 April 2024) [14,21]. The 3D structures of the S protein trimers of the BC8, BC18, LUN, and ENT strains were predicted using SWISS-MODEL (https://swissmodel.expasy.org/) and visualized using PyMOL2.6 software (https://pymol.org/2/, accessed on 25 April 2024). Phylogenetic analysis was performed using MEGA11 software (http://www.megasoftware.net/, accessed on 25 April 2024) with a bootstrap test of 1000 replicates.

### 2.10. Statical Analysis

Student’s t-test was conducted to compare the two BCoV strains’ infectious titers in PFU/mL at different time points and diameters of plaque sizes utilizing GraphPad Prism, version 9.0 (https://www.graphpad.com/, accessed on 25 April 2024). Significance levels were set at a *p* < 0.05.

### 2.11. Nucleotide Sequence Accession Numbers

The genomic information for strains BC8, BC9, BC18, BC39, and BC47 was submitted to GenBank (accession no. OR502440-OR502444).

## 3. Results

### 3.1. Isolation of Five BCoV Strains in HRT-18 Cells

For the isolation of BCoV, experiments were conducted using the HRT-18 cell line. Of the 15 samples that tested positive by RT-qPCR, five strains, namely, BC7, BC8, BC9, BC39, and BC47, were successfully isolated. BC7 was obtained from the same farm as BC8 and BC9; therefore, only BC8 and BC9 were sequenced as representatives at the second and third passages, respectively. Although BC18 was collected from a nasal swab and could not be isolated in HRT-18 cells, we were still able to obtain its whole genome sequence. BC8 and BC18 were paired samples collected from different tissues of one calf. It is important to note that not all isolates exhibited obvious CPE until the serial passages. Therefore, the confirmation of BCoV isolation was based on a combination of methods including RT-qPCR, RT-PCR, IFA, or IEM.

During the cell isolation procedure, two different conditions were performed: Inoc (−) condition and Inoc (+) condition, and BCoV strains showed similar growth patterns in these two conditions. BCoV antigens in cell culture were detected using anti-Mebus guinea pig hyperimmune antiserum NR455 (Figure 1). BCoV antigens were observed as green immunofluorescence, cell nuclei were stained with DAPI and shown in blue, the merged images of antigen and DAPI were obtained, and the CPEs were observed under light microscopy. From the IEM images, BCoV particles were observed to have obvious spike (S) protein projections and shorter hemagglutinin-esterase (HE) protein projections on the surface (Figure 2).

### 3.2. Growth Kinetics and Plaque Morphology Comparison between Current BC8 Strain and Historical DBA Strain

To assess virological differences between the current BC8 strain and the historical DBA/1990 strain, one final multi-step growth kinetics experiment was conducted at a multiplicity of infection (MOI) of 0.01 for both viruses (Figure 3A). The replication kinetics of the BC8 and DBA strains demonstrated peak titers at 48 hpi (DBA-P5 and BC8-P8). The BC8 strain reached 7.06 ± 0.21 log_10_ PFU/mL, and DBA reached 6.85 ± 0.00 log_10_ PFU/mL at this time point. Notably, the peak titer of the BC8 strain was significantly higher than that of the DBA strain (*p* < 0.01).

The plaque morphology of the BC8 and DBA strains exhibited noticeable differences in size (Figure 3B,C). The average plaque size of BC8 was measured to be 1.72 ± 0.39 mm in diameter, while the average plaque size of DBA was slightly larger at 1.99 ± 0.34 mm, based on measurements of 30 plaques (Figure 3D). Importantly, the plaque size of DBA was significantly larger than that of BC8 (*p* < 0.01).

### 3.3. Amino Acid Analysis of the S and HE Proteins between Respiratory and Enteric Strains

BC8 and BC18 were collected from the fecal sample and nasal swab, respectively, from the same calf. Therefore, the S protein sequences of BC8-P2 and BC18 were compared. We identified two amino acid differences at positions I617T and P960L. Amino acid changes in the S protein of these new strains and three pairs of historical respiratory and enteric strains are summarized in Table 2. To better show the sequence comparison between each pair of enteric and respiratory samples from one animal, the data are also shown in Appendix A. Based on this information, we speculated that some residues may contribute to BCoV respiratory and enteric tropisms and disease types. In residue 1180, three of four R strains had glycine (G, a zwitterion amino acid) instead of aspartic acid (D, a negatively charged amino acid) for all E strains and one R strain. Other amino acid variable sites between E and R strains were located at residues 45, 113, 370, 483, 531, 617, 743, 754, 960, 1052, 1196, and 1242. Among them, residues N45K, Y370D, S483P, D531N, I617T, S743I, P960L, A1052T, and Y1242D were non-conservative mutations and may have contributed to the different E and R tropisms.

Similarly, we compared the HE amino acid sequences of BCoVs. The five BCoVs identified in this study did not contain the 4-aa-insertion or 4-aa-deletion in HE proteins in some recently reported US BCoV strains [22]. No unique residues were found between the four pairs of respiratory and enteric BCoV strains.

### 3.4. D Structure Prediction of S Protein Trimer and Location of Three Amino Acids Potentially Related to Disease Types

We predicted the three-dimensional (3D) structure of the S protein trimer for BC8 and BC18 using protein structure prediction software online (Figure 4). Like other coronaviruses, the BCoV S protein trimer is composed of three protein monomers: chain A, chain B, and chain C, depicted in magenta, cyan, and green, respectively. At amino acid site 617, located in the C-terminal domain of the S1 subunit (Figure 5), BC8 had Iso^617^, while BC18 exhibited Thr^617^. More detailed stick structures are shown in red and yellow for amino acid residue 617 for BC8 and BC18, respectively (Figure 4A). At amino acid site 960, located in the S2 subunit and near the fusion peptide region, BC8 displayed Pro^960^, whereas BC18 displayed Leu^960^. More detailed stick structures are depicted in brown and purple for amino acid residue 960 for BC8 and BC18, respectively (Figure 4B). Residue 1180 was located between heptad repeat 1 and heptad repeat 2 regions (Figure 5), with Asp^1180^ in the BC8 strain and Gly^1180^ in the three respiratory strains (Figure 4C).

### 3.5. Phylogenetic Analysis of BCoV Isolates’ Whole Genome Sequence, S Protein, and HE Protein

To gain insights into the BCoV genetic diversity over the past few decades, we conducted phylogenetic analyses of the WGS, S proteins, and HE proteins. The WGS phylogenetic analysis included 21 historical BCoV strains and one human coronavirus OC43 strain as a control (Figure 6). The BCoV strains were primarily categorized into three genetic clades: (1) Historical clade, representing early classical strains such as Mebus and Quebec, (2) European clade, and (3) American–Asian clade. The new isolates were clustered within the American–Asian clade.

Phylogenetic analyses of the S proteins and HE proteins were conducted separately, including 33 BCoV strains with one control for S proteins and 28 BCoV strains with one control for HE proteins. From the phylogenetic trees, there was no clear clustering pattern observed for respiratory and enteric strains. While most respiratory or enteric strains formed small distinct groups, there were instances where they were intermingled with the opposite strains.

## 4. Discussion

It has been several decades since the discovery of the first BCoV Mebus strain in 1972 [23]. However, there remains a pressing need to investigate the evolution of American strains of BCoVs and isolate contemporary strains. BCoV causes both respiratory and enteric diseases, yet the underlying mechanisms remain poorly understood despite extensive research efforts on strain and disease characterization [14,21].

From our study, we obtained genome information for five current BCoV strains, shedding light on the current evolutionary status of BCoVs. We isolated five BCoV strains using HRT-18 cells. Interestingly, BCoV-infected cells exhibited minimal CPE until subjected to serial passages. For studies reliant on observing CPE, alternative cell types such as Vero or Madin–Darby bovine kidney (MDBK) cells might yield more pronounced effects [24,25]. In terms of RT-qPCR detection, although some samples showed low Ct values, corresponding to high viral RNA levels, the virus isolation and infectivity titers might not correspond well with mRNA levels. For instance, samples BC8 and BC18 exhibited Ct values of 35 and 22, respectively. However, only BC8, a fecal sample, could be successfully isolated in HRT-18 cells; BC18, a nasal swab sample, could not. This suggests that sample storage conditions or other factors may influence isolation success. Exploring alternative cell lines could also be beneficial. BCoV respiratory samples might replicate better in bovine/human respiratory epithelial cells, which mimic the viral natural growth environment. In this study, isolation experiments were conducted under two conditions: Inoc (+) and Inoc (−). The Inoc (+) condition retained the inoculum, probably containing bacterial metabolites, bile acids, and other factors from the original intestinal environment that could promote virus replication. In contrast, the Inoc (−) condition was performed to avoid potential toxic materials in the original sample that could be toxic to the cells and inhibit virus isolation. Unlike the porcine epidemic diarrhea virus [26], the five isolated BCoV strains exhibited similar replication under both conditions, although the inoculated cells in the Inoc (+) condition detached more quickly, which might have been due to the presence of large intestinal contents.

The detection of BCoV yielded a positive rate of 21% (15/71), indicating that BCoV remains circulating and represents a risk to herds’ health. This underscores the continued importance of addressing BCoV as a major infectious agent in farming operations. Investigating BCoV co-infections with other pathogens, such as *Cryptosporidium parvum*, rotavirus, *Salmonella enterica*, and *E. coli* K99, is essential for understanding the impact of BCoV co-infections on bovine health and farm biosafety. The S protein and HE protein play crucial roles in virus binding to host cell surface receptors [27,28]. BCoV utilizes N-acetyl-9-O-acetylneuraminic acid as a receptor, not only for the agglutination of erythrocytes but also for infecting cultured cells [29,30]. The S protein exhibited pairwise identities ranging from 90.15% to 99.85%, while the HE protein showed pairwise identities ranging from 93.40% to 100%. This indicates a higher degree of conservation in the HE protein compared with the S protein. These findings suggest a complex evolutionary history of BCoVs, with strains from different geographical regions and phenotypes showing mixed clustering patterns rather than distinct separation based on disease types.

In our virological experiments, we observed that the BC8 strain exhibited similar growth kinetics to DBA but had higher titers than DBA. However, despite its higher titers, BC8 had smaller plaque sizes than DBA. This suggested that viral titer may not necessarily correlate with the spread of a virus to neighboring cells, as factors such as cell–cell junctions, host cytoskeleton, and other unknown virus–host interaction factors also play a role in determining infectivity [31]. One interesting phenomenon observed in the plaque assays was that BCoVs did not exhibit high virulence sufficient to induce complete cell detachment and primarily caused cell damage, which may be the pathogenic characteristic of BCoVs.

Given that samples from different animals exhibited more variation, we focused on analyzing BC8 and BC18. They were both collected from the same calf, with BC8 coming from a fecal sample and BC18 from a nasal sample. The S protein acts as an important receptor-binding protein and contains a furin cleave site, facilitating membrane fusion and the release of the viral genome to the cytoplasm [32,33]. We noted that BCoV S proteins had the furin cleavage site 763KRRSRR768 (position based on BC8 S protein), which is related to viral infection and cell–cell fusion [34,35]. Amino acid residue 617, located in the S1 C-terminal domain, and residue 960, located in the S2 subunit, were different between BC8-P2 and wildtype BC18. These two residues may be related to BCoV respiratory and enteric tropisms or cell culture adaptation.

D1180 was consistently observed in the four enteric strains, while G1180 was predominant in respiratory strains (three of four) (Table 2 and Appendix A). The D1180G mutation lies between the HR1 and HR2 regions of the S2 subunit (Figure 5). Because the S1 but not the S2 subunit contains a receptor-binding domain, this D1180 mutation is less likely to lead to changes in tissue tropism. The D1180G mutation may simply provide an advantage for viral replication in the respiratory tract. Due to the limited number of paired spike sequences in this study, we cannot draw definitive conclusions. Further investigation using reverse genetics systems and animal studies is warranted to explore whether certain residues are related to BCoV replication efficiency in different tissues.

In summary, we successfully isolated five BCoV strains, BC7, BC8, BC9, BC39, and BC47, from feces and sequenced BC8, BC9, BC39, and BC47 and the BC18 respiratory strain directly from a nasal sample. Our findings contribute to the understanding of the current virological and sequence diversity of BCoVs. These findings offer valuable insights for future research endeavors aimed at elucidating BCoV pathogenesis, antigenicity, and tissue tropism. Furthermore, the data we have amassed lay the groundwork for the development of a BCoV reverse genetics platform, which promises to enhance our comprehension and management of this significant bovine pathogen. Besides this, we are currently constructing a BCoV reverse genetics platform that will enable us to edit viral genes and change targeted sites at the DNA level, a capability that has not been available for BCoV so far. The role of these predicted amino acid sites in BCoV tissue tropisms will be explored. This study was the foundation of our lab’s long-term BCoV studies.

## Figures and Tables

**Figure 1 viruses-16-00965-f001:**
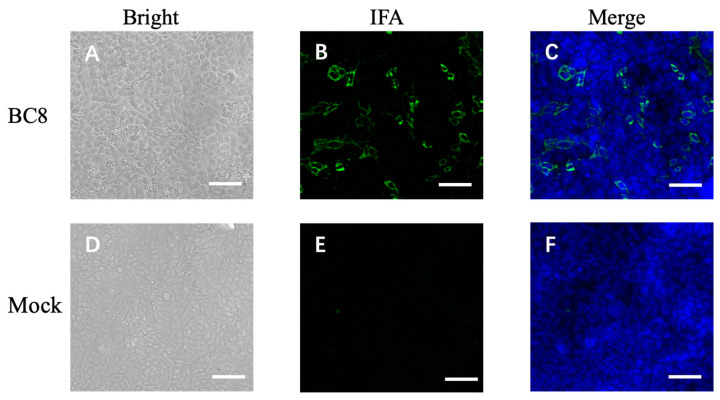
Immunofluorescent microscope image of uninfected HRT-18 cells (mock) and HRT-18 cells infected with the BCoV BC8 strain showing BCoV antigens in green color in the cytoplasm of infected cells and syncytia containing different numbers of nuclei. The BCoV antigens were detected by primary antiserum guinea pig hyperimmune serum raised against the BCoV Mebus strain (**B**). The blue is for nuclei staining with DAPI. The merged images are shown (**C**). The same BC8-infected cells under light microscopy, showing cytopathic effects (**A**). Uninfected HRT-18 cells under bright light (**D**), under immunofluorescent light (**E**), and merged with DAPI (**F**) are shown. Scale bar: 50 μm.

**Figure 2 viruses-16-00965-f002:**
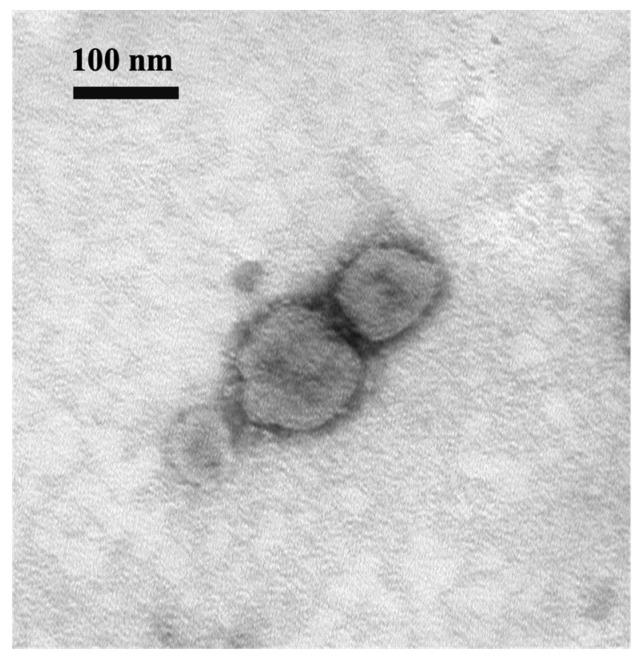
Electron micrograph of BCoV BC8 viral particles harvested from virus-inoculated HRT-18 cells. Scale bar: 100 nm. Magnification: 100 k. EM Model: Hitachi H-7500.

**Figure 3 viruses-16-00965-f003:**
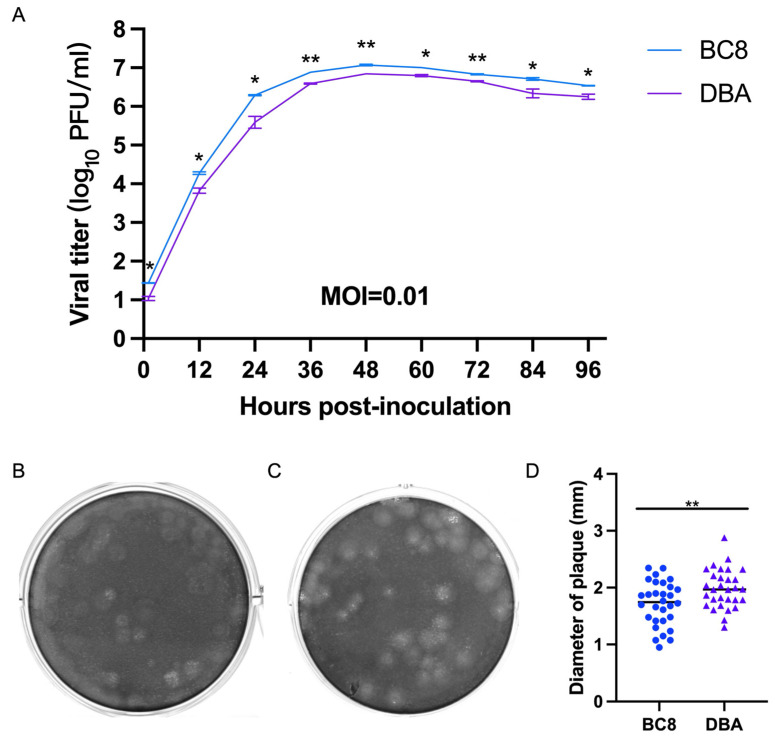
(**A**) Multi-step growth curves of the BCoV BC8 and DBA strains in HRT-18 cells. Infectious viral titers were shown at different time points (1 hpi, 12 hpi, 24 hpi, 36 hpi, 48 hpi, 60 hpi, 72 hpi, 84 hpi, and 96 hpi) post-inoculation for the BC8 or DBA strain with an MOI = 0.01. Each strain had duplicates at each time point. Plaques caused by BCoV BC8 (**B**) and DBA (**C**) at 72 hpi were shown (final agarose overlay concentration was 0.75%). (**D**) The plaque diameters of BC8 (1.72 ± 0.39 mm) and DBA (1.99 ± 0.34 mm) were measured at 72 hpi based on 30 plaques per virus (* *p* < 0.05, ** *p* < 0.01).

**Figure 4 viruses-16-00965-f004:**
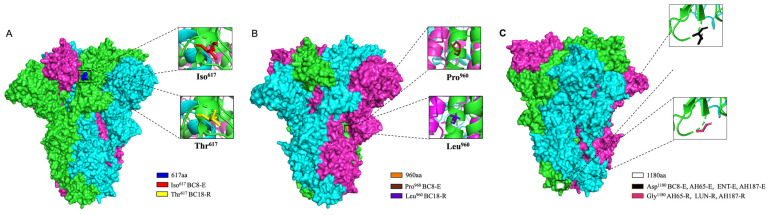
The 3D structural analyses of the predicted S protein trimers of BCoV strains using protein structure prediction software online (https://swissmodel.expasy.org/, accessed on 25 April 2024). The surface model used BC8 and LUN strains as templates and the stick model used all corresponding strains to predict. The location of amino acid residue 617 in the predicted S protein trimer is shown in blue and red in the detailed stick model for Iso^617^ in the BC8-E strain and yellow for Thr^617^ in the BC18-R strain (**A**). The location of amino acid residue 960 in the predicted S protein trimer is shown in orange, the detailed stick model for Pro^960^ in the BC8-E strain is shown in brown, and Leu^960^ in the BC18-R strain is shown in purple (**B**). The location of amino acid residue 1180 in the predicted S protein trimer is shown in white, the detailed stick model for Asp^1180^ in the BC8-E, AH65-E, ENT-E, and AH187-E strains is shown in black, and Gly^1180^ in the AH65-R, LUN-R, and AH187-R strains is shown in a warm pink color (**C**).

**Figure 5 viruses-16-00965-f005:**
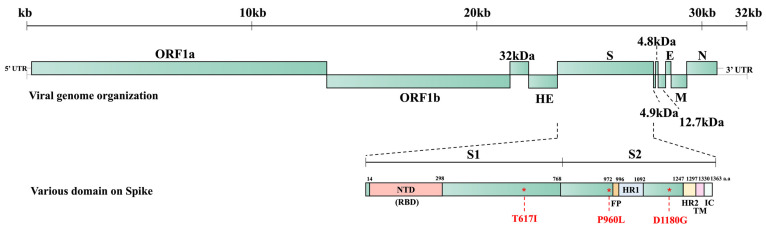
BCoV whole genome structure and the predicted domains of the S protein of BC8 strain. S1: S1 subunit, S2: S2 subunit, NTD: N-terminal domain, FP: fusion peptide, HR1: heptad repeat 1, HR2: heptad repeat 2, TM: transmembrane domain, IC: intracellular segment.

**Figure 6 viruses-16-00965-f006:**
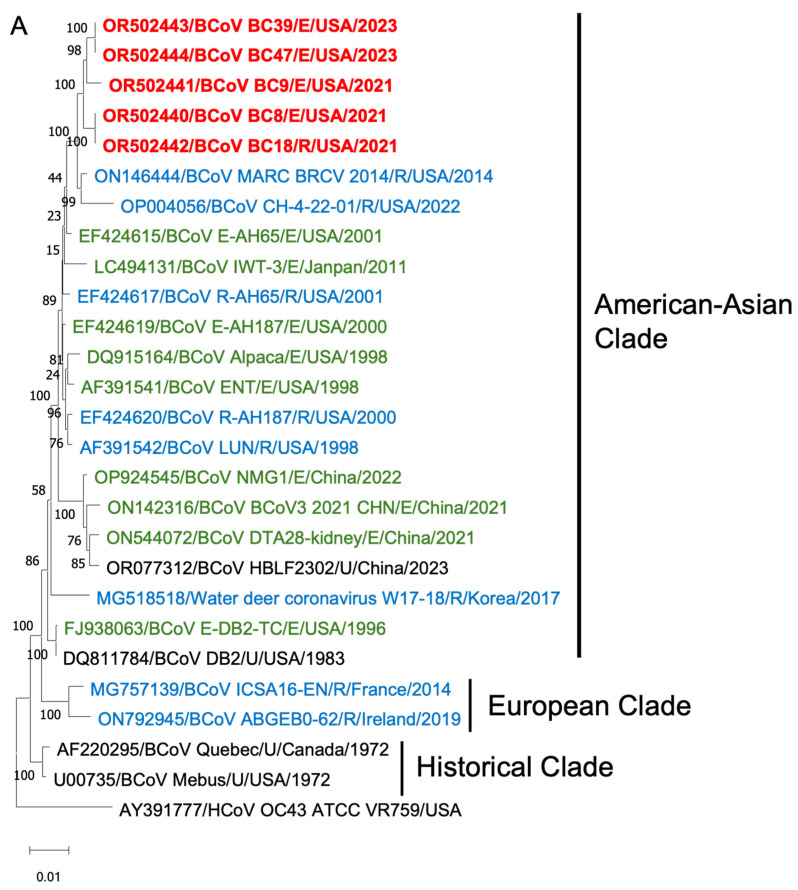
Phylogenetic analysis of BCoVs based on the entire genomic nucleotides (**A**), S protein amino acid sequences (**B**), and HE protein amino acid sequences (**C**) of five BCoV strains from this study (in red), historical BCoV strains, and the human coronavirus OC43 strain (out-group control). GenBank accession number, strain name, strain type, and detection country and year were indicated for each BCoV strain in the tree. Historical clade, European clade, and American–Asian clade are labeled. R: respiratory samples collected from nasal, lung, or other upper respiratory tissues (in blue); E: enteric samples collected from fecal or other enteric tissues (in green); U: unknown strain type (in black).

**Table 1 viruses-16-00965-t001:** Sample information and RT-qPCR results ^1^.

Farm	Breed	Age	Type	Positive Samples ^2^	Positive Rate (%)
OH	NA	Pre-weaned ^3^	Fecal	BC7, BC8, BC9, BC10	4/10 (40)
			Nasal	BC12, BC14, BC15, BC17, BC18, BC19	6/10 (60)
GA1	Holstein	Pre-weaned ^3^	Fecal	0	0/18 (0)
		Weaned	Fecal	BC33, BC39, BC47	3/11 (27)
	Jersey	Pre-weaned ^3^	Fecal	0	0/6 (0)
		Weaned	Fecal	BC37	1/8 (13)
GA2	Nigerian red	Pre-weaned ^4^	Fecal	BC53	1/3 (33)
		Weaned	Fecal	0	0/1 (0)
GA3	NA	Weaned	Fecal	0	0/3 (0)
	NA	Nursing beef	Fecal	0	0/1 (0)

^1^ RT-qPCR positive is defined as Ct < 35. ^2^ Underlined and bold samples were isolated successfully in HRT-18 cells. ^3^ These calves were separated from their mother with milk feeding. ^4^ These calves were separated from their mother with milk replacement feeding. NA: Not available.

**Table 2 viruses-16-00965-t002:** Analysis of the S protein amino acid variations between four pairs of respiratory strains and enteric strains (each pair of samples was from the same animal. E: strains from enteric samples, R: strain from respiratory samples).

Position in S Protein	Consensus Amino Acid	OR502440/BC8/E (Animal 1)	EF424615/E-AH65/E (Animal 2)	AF391541/ENT/E (Animal 3)	EF424619/AH187/E (Animal 4)	OR502442/BC18/R (Animal 1)	EF424617/R-AH65/R (Animal 2)	AF391542/LUN/R (Animal 3)	EF424620/AH187/R (Animal 4)
24	V	*	L	*	*	*	*	*	L
35	S	F	*	*	*	F	*	*	*
45	N	*	*	*	*	*	*	K	*
113	I	*	*	*	*	*	V	*	*
174	P	S	*	*	*	S	*	*	*
179	Q	R	*	*	*	R	*	R	*
370	D	*	*	Y	*	*	*	*	*
483	P	*	*	S	*	*	*	*	*
492	D	G	*	*	*	G	*	*	*
499	N	*	S	*	S	*	S	*	*
501	S	*	*	P	*	*	*	P	P
509	T	*	N	*	N	*	N	*	*
510	T	*	*	S	*	*	*	S	S
525	H	Y	*	*	*	Y	*	*	*
531	D	*	*	*	*	*	N	*	*
546	P	S	*	*	*	S	*	*	*
554	Y	H	*	*	*	H	*	*	*
571	H	*	Y	*	Y	*	*	*	Y
578	S	*	*	T	*	*	*	T	*
617	T	I	*	*	*	*	*	*	*
743	S	*	*	*	*	*	*	*	I
754	S	*	*	*	*	*	*	*	N
960	P	*	*	*	*	L	*	*	*
1052	A	*	*	*	*	*	*	T	*
1180	D	*	*	*	*	*	G	G	G
1192	N	Y	*	*	*	Y	*	*	*
1196	T	*	*	*	*	*	*	*	S
1232	M	K	*	*	*	K	*	*	*
1242	D	*	*	Y	*	*	*	*	*

* The same amino acid as the consensus sequence.

## Data Availability

The data presented in this study are available in this article and on request from the corresponding author.

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
