# Peer review of "Isolation and Characterization of Contemporary Bovine Coronavirus Strains"

_viruses, 2024, doi:10.3390/v16060965_

Round 1

Reviewer 1 Report

Comments and Suggestions for Authors

This is an important study in evaluating the various types of BCoV and comparing them to historic strains. 

I am not qualified to evaluate the description of the molecular methods. However, I have concerns regarding that you are using a haphazard number of strains, that are not well described and thereafter to talk about associations with clinical outcomes. Your study does not have the design or power to evaluate relationship between BCoV finding and clinical outcomes. I would remove all these components and focus on variability between isolates within an animal (respiratory vs enteric strain) or between animals. The statistical analysis section mention that you use Anova, but I do not see where this is used in the results section. Table 2 is not very clear to readers. 

Line15. Abstract. I recommend writing in neutral and not first person (unless you are writing in the discussion).

Line 23. All these strains from ‘a cattle’ or from several cattle?

Line 42. ‘open reading’

Line 78 and Line81. From how many farms were these isolates?

Line 241. Expand the statistical analysis section. Indicate what parameters were tested against what outcomes, and the samples used. Were any repeated measures or random effects of farm included in the analysis.

Line 253. This is the first time that you reveal what information that was collected for the sample, such as clinical diagnosis. This should be stated in the material and methods.

Line 257. How precise was the age determination of the calves sampled. Were they on milk or recently weaned. The presence of clinical signs in relation to BCoV positive findings is strongly linked to the age and stress status of the calf.

You are working with  an extremely random sample set to determine associations with clinical outcomes in the animals. Also, the presence of diarrhoea in the young stock is strongly linked to farm sampled and concurrent other disease pathogens present on the farm.

Table 2. This table is not easy to understand for the reader. What conclusions do you want the reader to make from this table? If you use * to indicate a footnote, then you should not use * to mark other things in the table.

Where is the statistical analysis in the results?

Line 419. Since the reader knows little about the number of farms sampled, or the exact age of calves at sampling it is difficult to evaluate what a 9% recovery rate is. Furthermore, this should not be compared to the 10 diarrhoeic disease samples from Georgia. Or were these samples part of that same dataset?

Reviewer 2 Report

Comments and Suggestions for Authors

Coronaviruses (CoVs) infect a wide range of mammalian (including human) or avian species and may cause life threatening diseases. CoVs infecting livestock can lead to huge economic losses, highlighting the relevance of their surveillance and control. In this line, the manuscript is focused in bovine CoV (BCoV), which is an understudied animal CoV, when compared with other CoVs infecting farm animals. The manuscript describes the isolation and basic characterization of currently circulating BCoV viruses. The manuscript is well-written, the rationale is clear, and the experiments are justified. However, there are a few issues that would be considered to improve the manuscript.

Specific comments:

1. As pointed out by the authors, tissue culture adaptation cannot be excluded, and maybe the use of different cell lines representing different species and/or tissues would be relevant. Related with this comment, although low-passage viruses (P2 or P3) were used for NGS, it would be interesting to compare these sequences with those obtained from the original samples from the animals (at least in selected cases). Was that possible? Alternatively, it would be interesting to demonstrate the infectious ability in vivo (i.e., tissue tropism) of the isolated viruses. These issues would at least be discussed.

2. Samples BC8 and BC18, from the same calf are very interesting, and one of the strongest points in this manuscript. They may represent the intrahost diversity of the virus, and may give clues on how BCoV infection progresses into the host. It would be worthy to try isolation of BC18 in respiratory cells. Related with this comment, in case BC18 can be isolated, can BC8 infect bovine respiratory cells or BC18 infect bovine enteric cells?

3. Related with previous comments. As no virus could be isolated from the nasal samples (in some cases despite an elevated Ct value), how can BCoV tropism be confirmed? Is it possible that the viruses in the nasal tissue could be non-infectious “sub-products” of the enteric infectious virus?

4. Lines 321-323 and 458-464. The comparison of SARS-CoV-2 S D614G mutation with BCoV S D1180G seems misleading. Please note that the relevance is not in the amino acid change (D by G, caused by a transition A by G that is more frequent than transversion mutations), but in the S protein domain where it is located. In this sense, position 614 is in S1 domain, while 1180 is in S2 domain, therefore, the consequences of these mutations may be completely different.

Minor comments:

1. Lines 91-92. Please note that the RT-qPCR assay is designed to amplify a sequence into M gene. Therefore, although used for diagnosis, it would be not appropriate to measure viral gRNA (as both gRNA and several sgmRNAs are detected with this assay) or viral titers.

2. Lines 370-372. Please note that the information about GIa and GIb in this text is opposite to what is shown in Fig. 6. Please, correct where it is wrong.

3. Line 252. Please note that if it is “among the positive samples” then “10/166 (6.02%)” should be replaced by “10/16 (62.5%)”

4. Page 2 Line 86. Please note that Table 1 is first mentioned here. However, the table is located in Page 6. Maybe Table 1 location in the text would be re-considered

Round 2

Reviewer 1 Report

Comments and Suggestions for Authors

This paper now has a better focus on your aims and intentions. You have addresed previous expressed concerns. I am interested in your tissue tropism theory. It needs to be investigated further, as when I do large scale sampling of calves for BCoV, I find that the majority of calves are simultaneously excreting BCoV in fecal and nasal samples. 

I attach some minor comments. 

Author Response

Reviewer 1

This paper now has a better focus on your aims and intentions. You have addressed previous expressed concerns. I am interested in your tissue tropism theory. It needs to be investigated further, as when I do large scale sampling of calves for BCoV, I find that the majority of calves are simultaneously excreting BCoV in fecal and nasal samples.

Answers: Thank you for your comments. We plan to use reverse genetics system to investigate viral mutations and tissue tropisms.

Minor comments:

Line 78. I do not understand if the calves were weaned, it means that they are no longer provided milk, but you mention that they are supplemented with milk.

Answers: In dairy and veal farms, the newborn calves were separated from their mother immediately or within a few days after birth and raised by artificial milk feeding. So, we replaced "weaned" in the text (line 78) and the footnotes of Table 1 to " separated from their mother " to clearly describe the calf age and feeding conditions.

Table 1. I do not know what you mean with clinical rate %. I suggest you remove this, unless you explain in introduction that you collected information on the health status of the sampled calves at the time of sampling.

Answers: As suggested, we have removed the clinical rates from Table 1.

Line 250. In the section Statistical analysis I would like to see what associations were evaluated statistically with the student t-test. Thus what variables were compared, such as plaque forming units at peak titres etc.

Answers: The text was revised to "Student’s T-test was conducted to compare the two BCoV strains' infectious titers in PFU/mL at different time points and diameters of plaque sizes……" (line 253-254).

Line 288. I do not see how many of these kinetics experiments were run. I notice that you

mention 24 well plates, but not sure what sample size you are having here. Line 290, thus these experiments (that you state are in plural), need to spell out how many.

Answers: One final growth kinetics experiment was run in 24-well plates, with duplicates per timepoint (Fig. 3A). Another preliminary growth kinetics experiment was performed. The text and Fig. 3A have been revised accordingly.

Line 296, it is recommended to write out the exact p-value to two decimals and not just p < 0.05.

Line 301. This is correct p-value specification, when it gets less than 0.01, then p < 0.01 is

correct.

Answers: The exact p value is 0.0042 for the peak titers at 48 hpi. So, the text was revised to p < 0.01 (line 296). We double checked other timepoints and the p values for 36 hpi and 72 hpi were 0.0015 and 0.0079, respectively. So, the "*" for 36 hpi, 48 hpi and 72 hpi in Fig. 3A has been changed to "**".

Table 2. Just a suggestion. I recommend putting a column under the title column indicating the animal, as not all people are using colour printouts. When searching for intra-calf variability, the table may be easier to read when the enteric and the fecal isolate from the same calf are standing adjacent in the table.

Answers: We added Animal 1, Animal 2, Animal 3, and Animal 4 in the title row to indicate that the four pairs of samples were from four cattle. We listed the enteric and respiratory samples adjacent to each other, respectively, to better show the potential patterns between the E and R strains.

Figure 4. Small grammatic comment. You can write this is present tense, you write the colours were shown… well they are shown.

Answers: The sentences were revised as suggested.

Line 393. Remove the word ‘circulating’, as you do not know this.

Answers: The word "circulating" is removed.

Line 394. Remove the word ‘Meanwhile’

Answers: The word "Meanwhile" is removed.

Line 417. Remove this sentence, there is absolutely nothing new here, we do not know from

where these samples came, and the recovery of these fecal pathogens is totally normal and expected. You can instead elaborate the following sentence such as ‘Investigating BCoV co-infections with other common enteric pathogen such as Cryptosporidium parvum, rotavirus, Salmonella enterica, E. coli K99, …. Note that Salmonella enterica and Cryptosporidium parvum, the genus is capitalized and the genus and species are in italics.

Answers: The sentence was removed and the next sentence was revised to "This underscores the continued importance of addressing BCoV as a major infectious agent in farming operations. Investigating BCoV co-infections with other pathogens, such as Cryptosporidium parvum, rotavirus, Salmonella enterica and E. coli K99, is essential for understanding the impact of BCoV co-infections on bovine health and farm biosafety." (lines 422-423).

Line 462. This is interesting. I am currently performing a study where I find strains

simultaneously in about 60% of calves sampled (nasal and fecal samples taken

simultaneously). This proposed tissue tropism needs to be explored further.

Answers: Thank you for sharing this information. I would like to discuss with you further and hope perform sequence analyses for the paired samples from one animal when their genomic sequences are available. We plan to use reverse genetics system to investigate viral mutations and tissue tropisms.